# Patient and practitioner perspectives on the design of a simulated affective touch device to reduce procedural anxiety associated with radiotherapy: a qualitative study

Hugo Hall ![ORCID],[1,2] Yasmin Dhuga,[2,3] Caroline Yan Zheng,[4] Gemma Clunie,[5,6] Elizabeth Joyce,[7] Helen McNair,[7,8] Alison H McGregor[5]

HH and YD contributed equally.

For numbered affiliations see end of article.

**Correspondence to**
Professor Alison H McGregor;
a.mcgregor@imperial.ac.uk

## ABSTRACT

**Objective** The aim of this study was to elicit the views of relevant stakeholders on the design of a device using simulated affective touch to reduce procedural anxiety surrounding radiotherapy and imaging.

**Design** This qualitative study collected data from focus groups which were then analysed using inductive thematic analysis in line with Braun and Clarke's methods.

**Participants and setting** Twenty patients and carers were recruited, as well as 10 healthcare practitioners involved in either delivering radiotherapy or imaging procedures.

**Results** Patients, carers and healthcare practitioners agreed on some aspects of the device design, such as ensuring the device is warm and flexible in where it can be used on the body. However, patient and healthcare practitioner cohorts had at times differing viewpoints. For example, healthcare practitioners provided professional perspectives and required easy cleaning of the device. Meanwhile patients focused on anxiety-relieving factors, such as the tactile sensation of the device being either a vibration or pulsation. There was no consensus on who should control the device.

**Conclusions** The desired features of a simulated affective touch device have been investigated. Different priorities of patients and their carers and healthcare practitioners were evident. Any design must incorporate such features as to appease both groups. Areas where no consensus was reached could be further explored, alongside including further patient and public involvement in the form of a project advisory group.

## Strengths and limitations of this study

► This is the first qualitative study to explore whether a tactile intervention provides a useful tool in reducing procedural anxiety for radiotherapy patients.
► This study explores the opinions of two separate stakeholder cohorts, each with different viewpoints (personal and professional) around the design of the device.
► All participants were selected from a single geographical area which may indicate a homogeneous focus group composition.
► Our study cannot be generalised to all individuals who receive radiotherapy, as this study only considered adult participants.

## INTRODUCTION

Currently 458 000 people live with cancer in the UK.[1] Over 50% of these patients will be managed with radiotherapy,[2,3] which can cause procedural anxiety, with approximately 49% of patients experiencing anxiety and psychological distress.[4,5] Procedural anxiety refers to excessive worry or fear of medical procedures[6] and is a phenomenon exacerbated by new developments in radiotherapy, for example, stereotactic and adaptive radiotherapy, that require long treatment times.

High levels of distress during radiotherapy can directly impact the accuracy and efficacy of the procedure.[7] Procedural anxiety is not limited to radiotherapy and also occurs during diagnostic imaging, and other invasive procedures performed on a conscious patient. Between 2% and 5% of MRI scans are terminated due to procedural anxiety[8,9]; this equates to a significant financial cost for the National Health Service (NHS) considering that 3.4 million MRI scans were conducted between 2017 and 2018.[10]

Pre-existing interventions to manage procedural anxiety are limited and there is a need for an appropriate anxiety-alleviating strategy or device in medical settings.[11–13] Techniques such as verbal communication or music may distract the patient but do not address the reported sense of isolation associated with separation from caregivers or families. Sedation is commonly used to tackle procedural anxiety in the context of MRI but has

implications for cost, risk and time.[9] Sedation, especially if treatment is required on a daily basis, has implications on service provision, facility usage and hospital resources, and repeated sedation may have associated medical risks in some patient groups.

Research indicates that human affective (or empathetic) touch 'provides both psychological and physical comfort'.[14] However, relatives and carers are prohibited from being in the radiotherapy treatment room, leaving patients alone. Since the COVID-19 pandemic, patients are even more isolated with many hospitals not allowing visitors into waiting areas. As such, this research focuses on patients' perspectives of a simulated affective touch (SAT) device—a haptic device that uses tactile simulations to mimic attributes of human affective touch.

Haptic devices simulate attributes of human affective touch, such as gentle stroking, which has been shown to produce pleasant sensation,[15] relieve psychological distress[16] and reduce the sense of social isolation associated with being apart from caregivers.[17–19] However, such tactile interventions have not been widely explored as a tool to reduce procedural anxiety in radiotherapy patients. It is postulated that SAT can provide a non-invasive intervention to combat procedural anxiety surrounding radiotherapy due to their ability to reproduce natural touch.[20]

The following paper aims to investigate patient, carer and healthcare professional (HCP) views on the creation of a calming device using SAT to combat induced procedural anxiety. Patients were asked about what aspects of any SAT device they would find calming. Carers play a vital role in supporting their loved ones through difficult times, a view endorsed by a recent study (MacMillan survey),[21] which reported 74% of carers supported their loved ones by talking and listening to them. Therefore, we can assume they have insight on the impact of medical procedures on their loved ones and also have a role in the requirements of such a device. Finally, the views of HCPs as key stakeholders in any potential device were collected to ensure future clinical usability and translation. HCP views were collected in order to advise on professional requirements (eg, cleanability), as well as what they thought patients might appreciate about any calming SAT device. We intend to use the insights of the above groups to facilitate the future development of a design brief for an SAT prototype device to reduce procedural anxiety. This will ensure that the prototype device will be relevant and suitable for future clinical settings.

## METHODS
### Study design
This was a qualitative co-design study using focus groups to elicit the views of two distinct categories of participants (first patients and carers, second HCPs), on the creation of an SAT device for use during radiotherapy interventions.

The purpose of the focus group was to establish participant thoughts around the experience of four soft robotic

tactile intervention probing artefacts, which all had different shapes, touch patterns and interaction options. First, we established the ground rules and agenda of the focus group. Then, the four probing artefacts were introduced to the group to serve as prompts. This enabled participants to engage in discussion around the desirable and undesirable features of a tactile intervention device.

Example questions included:
1. How participants feel about the sensations.
2. Whether they would consider the device useful in helping to alleviate possible anxiety during the care journey.
3. Where during the care journey they think this device would be useful.
4. What sort of sensations they liked and would prefer.

We adopted a phenomenological approach using inductive thematic analysis. This allowed us to focus on how best to use the product design to counter the phenomena of procedural anxiety around radiotherapy and imaging.

### Participants and setting
Participants were recruited from two groups of stakeholders: one comprised both patients and carers, the other group comprised HCPs. Patients and carers were invited to participate as experts on the lived experience of having radiotherapy and/or imaging procedures; this allowed us to explore circumstances and situations that may have created anxiety and to provide ideas on how to manage these. The views of HCPs involved in delivering radiotherapy or imaging procedures were important as key stakeholders if the device is to become part of clinical practice.

Convenience sampling was used to select both participant groups. Inclusion criteria stipulated participants were aged over 18 years, while those patients who were considered too unwell (by self-report or in consultation with clinical staff) were excluded. Those who were unable to understand English without an interpreter were also excluded. Patients and their carers were contacted using flyers placed in radiotherapy treatment and pretreatment waiting rooms, while HCPs were internally recruited by departmental email. Patients from the local Biomedical Research Centre Patient and Public Involvement (PPI) groups were also emailed. Information sheets were distributed to interested patients and carers who then gave written informed consent to join the study. Written informed consent was also obtained from HCPs participating in the study.

### Data collection
A total of six focus groups took place at the Royal Marsden Hospital. The total number of participants was 30 (table 1), with four patient and caregiver groups (two groups with four participants, two groups with six participants) and two HCP groups (five participants each). Of the 10 HCPs, 9 were therapeutic radiographers (eight involved in the planning or treatment of radiotherapy patients and one educational lead) and 1 was a diagnostic

**Table 1** Description of focus groups and participants

|  | Patient had experience of radiotherapy | Cared for someone with radiotherapy experience | Other* | Worked as a healthcare practitioner in radiotherapy |
|---|---|---|---|---|
| Totals | 13 | 3 | 4 | 10 |
| Gender |  |  |  |  |
| Male | 4 | 1 | 3 | – |
| Female | 9 | 2 | 1 | 10 |
| Age |  |  |  |  |
| 18–25 | – | – |  | 1 |
| 26–35 | – | – |  | 5 |
| 36–45 | – | – |  | 2 |
| 46–55 | 2 | – |  | 2 |
| 56–65 | 7 | 1 | 2 | – |
| 66–75 | 3 | 2 | 2 | – |
| 76–85 | 1 | – |  | – |

A total of 30 participants were recruited: 13 radiotherapy patients, 10 healthcare practitioners and 3 carers. Four other participants had not received radiotherapy or cared for someone who had experienced radiotherapy.
*Other' refers to patients with cancer recruited from the BRC patient and public involvement group who had received imaging (eg, MRI) but not radiotherapy.
BRC, Biomedical Research Centre.

radiographer working mainly in MRI. The patient and caregiver focus groups lasted on average 83 min (ranging from 82 to 85 min). The HCP focus groups lasted on average 49 min (ranging from 48 to 50 min). Recruitment was stopped after six focus groups when initial analysis of the themes demonstrated no new emerging themes, indicative of saturation.[22]

Focus groups were recorded and transcribed verbatim for analysis. Each focus group was conducted by two medical facilitators with qualitative cancer research experience and were supported by the design researcher. All qualitative researchers involved in leading the focus groups had no prior relationship with the patient participants but two of them were well known to the HCP cohort. The varied comments and depth of discussion imply that that there was sufficient freedom for articulating thoughts in each of the groups. The design researcher was not known to any participants.

## Data analysis

Thematic analysis was used to identify themes and subthemes inductively within the focus group data. In line with Braun and Clarke's[23] methods, the initial step was for researchers to familiarise themselves with the interview recordings. Next, transcripts were analysed using QSR NVivo Release V.1.1 and all information relating to calming measures was coded. Each transcript was double coded (once by the design researcher and once by the medical researchers) allowing for multiple perspectives on the analysis. An independent researcher then reviewed the codes to increase intercoder reliability and the trustworthiness of the results.[24] Periodical meetings about the analysis with the whole project team led to iterations and finally the collation of the codes into themes and subthemes.

## Patient and public involvement

The concept was presented at initial planning stages to two Royal Marsden NHS PPI groups. Positive feedback was received which enabled us to proceed with the study.

## RESULTS

Three overarching themes emerged from analysis of the data: patient and carer views on what they want from the device, HCP views on the device and HCP views on what they believe patients want from the device.

Further analysis of the data identified seven key subthemes based on what patients wanted from the device: control of the device, temperature of the device, cleanliness of the device, where the device should be located on the body, shape of the device, visual appearance of the device and tactile sensation of the device.

There were eight key subthemes based on what HCPs believe patients would like: non-clinical appearance of the device, fitting the device into an object found in the treatment room, who controls the device, where the device should be located on the body, cleanliness of the device, temperature of the device, when to use the device and customisation of the device. These themes are summarised in figure 1 and discussed in more detail below.

## PATIENT AND CARER VIEWS
### Control of the device

Patients and carers reached strong consensus in favour of the patient being able to control the device themselves. They highlighted that in times of uncertainty, the device would give the patient something to be in control of. Another popular opinion included a relative being able

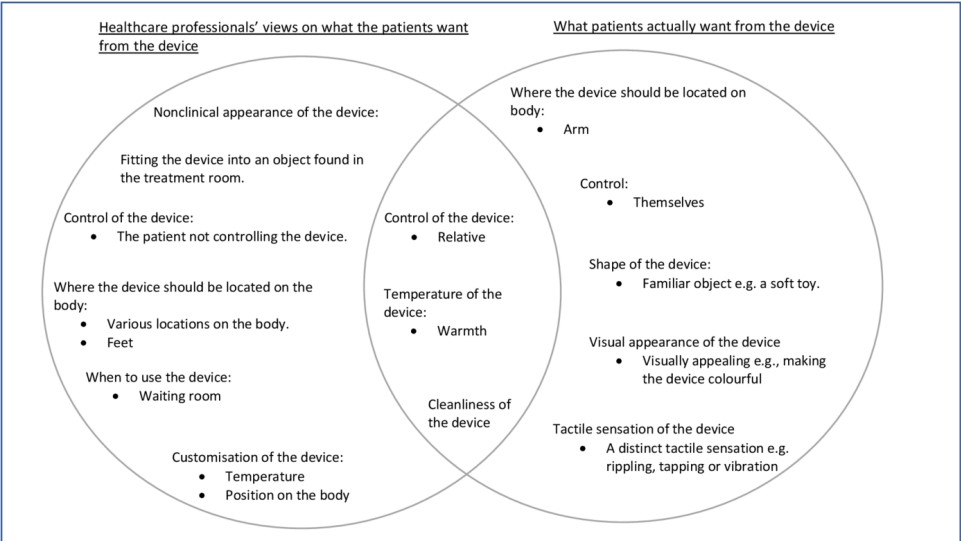

**Figure 1** An outline of the overarching themes and the subsequent subthemes within. The Venn diagram demonstrates where the patients and healthcare professionals agree and disagree with each other.

to control the device from home as many liked the idea of being able to interact with a loved one at a distance.

> That will make a big difference. And you've no control in these situations, so I think if you being able to control it probably would be quite reassuring. (P7: cared for someone with radiotherapy experience)

### Temperature of the device
The most discussed feature of the device, and the most important feature for some patients and carers, included the temperature of the device. The consensus was that the device should provide the patient with warmth.

> It's also temperature, when you hold someone's hand, it's a warm experience, whereas sometimes with some of the treatments you have, you feel quite cold and distant, and that's a bit of a strange one. It's like saying to someone it was all warm and fuzzy, it's a stupid thing to say, but actually temperature and touch and a feeling of wellbeing all go together, really. (P17: patient had experience of radiotherapy)

### Cleanliness of the device
Only one patient expressed a desire for a clean device and stated that they would not want to use a device shaped like a glove if someone else had used it.

> …but then hygiene wise. If someone has used the glove, I wouldn't want to use it afterwards. But that's me. (P7: cared for someone with radiotherapy experience)

### Where the device should be located on the body
Patient and carer preferences for location on the body included a wide variety of locations: the hand, arm, shoulder, neck, head, leg, foot, buttocks, whole body and customisable—being able to use the same device in

multiple locations. However, the most popular answer was using the device around the arm either because of a medical reason or personal preference.

> Maybe, for me, the arms, because I've got lymphedema, and that can be annoying. (P10: other (imaging but not radiotherapy experience))

### Shape of the device
Patients and carers expressed interest in the shape of the device. The ball-shaped probing artefact received good feedback with several participants expressing positive sentiments towards it based on its shape. Other participants suggested the shape of a teddy bear or comforting soft toy to help make associations with happier times.

> The ball, it would just give you a degree of confidence as someone who isn't so secure in where they are. (P19: patient had experience of radiotherapy)

### Visual appearance of the device
Patients and carers discussed making the device visually appealing so that it looked more interesting.

> Would it be possible to put lights or something inside it, so that it comes on… where the bits come out, colourful as well? (P12: patient had experience of radiotherapy)

### Tactile sensation of the device
Overall, patients and carers liked the idea of the device having a distinct tactile sensation, whether it was a pulsation, ripple, tapping or vibration. These motions were described as comforting and a form of distraction.

> It's clever isn't it? If you get it in the right pulse, it's just so calming. It is the ripple up and the ripple

down, modulating. (P3: patient had experience of radiotherapy)

## HEALTHCARE PRACTITIONER VIEWS
### Non-clinical appearance of the device
All HCPs expressed a negative attitude towards the device appearing like a piece of clinical equipment. This was due to their belief that this would be off-putting to the patient. Two HCPs expressed that they felt blue was the correct colour for the device from a personal perspective.

> You wouldn't want it to look too clinical. (P27: radiographer)

> Because of the patient age group. They might think, oh, you're taking my blood pressure. (P29: radiographer)

### Fitting the device into an object found in the treatment room
The HCPs believed that patients would favour a device that was prefitted into object found in the treatment room. They believed this would benefit patients who were nervous about disclosing their anxiety. From a professional point of view, the HCPs also believed that a piece of equipment designed to be left in situ would be more efficient.

> You don't actually have to ask for something. You don't even have to say yes, I'd like something for my anxiety. (P23: radiographer)

### Control of the device
The HCPs had different ideas about who the patient would most appreciate being in control of the device. Some HCPs believed that family members controlling the device would be the most comforting; enabling the patient to control the device could introduce a new anxiety on top of radiotherapy. Additionally, from a professional point of view, HCPs were concerned about giving patients control of the device. They were worried that the device could distract patients and affect their ability to remain still for the radiotherapy treatment or imaging.

> Yes, definitely that element of someone else can control it is great. And I think, actually, a lot of patients would find that really comforting to know that their family, I don't know, their children, or something, was doing that I think. (P26: radiographer)

> I just wonder if [controlling the device] introduces too many elements… They're overwhelmed already… He's relaxed, then he switches it off, then in treatment… They then panic because they want it back on. I think it introduces too many overwhelming factors. (P28: radiographer)

### Where the device should be located on the body
One of the most widely held views among HCPs was that patients would want a flexible device that could be used

in various locations on the body. If the device was not adaptable in terms of location, the second most favourable solution was to design a device for use on the feet. This was suggested both because of its similarity to a foot massage as well as because this is a location that was judged to be compatible with many radiotherapy set-ups.

> Maybe the ability for it to go… Adaptability, maybe, of different parts of the body or different equipment. Adaptability. (P28: radiographer)

### Cleanliness of the device
HCPs were keen to have a device that was easily cleanable and wipeable, with one group listing it as their top feature required when considering the practicality of an SAT device for use in radiotherapy. This research was conducted pre-COVID-19 pandemic, and so this feature would likely be of even greater importance now that infection control procedures and the disinfection of all equipment post-use have been brought into such sharp focus.

> Everything would need to be wipeable. (P27: radiographer)

> It has to pass infection control. (P25: radiographer)

> Facilitator: Practicalities. Where would you fit the practicalities?

> P27: I would say that it's cleanable being up at the top.

> P28: It has to be top, yes… (P27 and P28: both radiographers)

### Temperature of the device
In a point of similarity to the patients and carers, most HCPs believed that a warm device would be calming to a patient. However, a few HCPs disagreed and suggested that it would be good to adjust the temperature based on the needs of the patient.

> And, for me, I always like to make sure they're warm. Our treatment rooms can be quite cold. And if somebody is anxious, I don't know, I always think if I'm cold I'll be more anxious. So, it might be getting them an extra blanket, or making sure they've got an extra gown or something. (P25: radiographer)

### When to use the device
Some HCPs believed patients would want to use the device in the waiting room to alleviate anxiety.

> And that's I think one of the first points where you can really tackle that anxiety as such and alleviate some of it. (P28: radiographer)

### Customisation of the device
Overall, there was agreement on the fact that the device should be customisable in various ways in order to meet everyone's preferences. For example, patients should be able to choose whether they want to be able to feel the

warmth, the tactile sensation employed and the position on the body the device is used.

## DISCUSSION

Receiving cancer treatment can be an anxiety-provoking experience for patients for many reasons, including separation from their loved ones, an issue that is currently very pertinent.[4] With recent advances in radiotherapy, patients are required to spend an increasing amount of time in the treatment room, which may exacerbate such problems. Anxious patients may be managed pharmacologically but this is not appropriate for all patients and can be time-consuming. On the non-pharmacological side, there are no universally used interventions in radiotherapy departments, and research into new interventions is limited.[25] Further, any non-pharmacological interventions currently being investigated do not address the lack of physical interaction experienced by patients, instead focusing on distracting patients (for example, through music).

We have identified key design aspects for an SAT device that may alleviate the anxious patients' experience during radiotherapy and/or imaging. The implementation of such a device may help reduce costs through a smaller number of cancelled procedures as well as less spent on medication to combat procedural anxiety.

Overall, we found that participant cohorts had mixed opinions on the design aspects of such a device. This is not unexpected as HCPs have different roles and priorities to the patient and their carer.[26] Perhaps unsurprisingly, HCPs were concerned about the device not disrupting their workflow, for example, emphasis was placed on designing an easily cleanable device. This is consistent with the literature which suggests interruptions to their workload may contribute to medical errors.[27]

Control of the device was a controversial topic. Patients expressed the desire to have control of the device, whereas HCPs were keen for patients not to have control. HCPs feared that patient control would lead to disruptions to the treatment session. Both groups agreed that allowing relatives some degree of control of the device would be good. Future iterations of the device need to provide the level of control required by the patients while addressing HCPs' concerns in relation to service delivery. If the efficacy of the device is established, this may facilitate greater acceptance by the HCPs.

Future research should also incorporate further PPI and engagement, perhaps in the form of a project advisory group. Involving such an advisory group at all stages of the research could help to make sure the patient perspective is not neglected,[28] ensuring any device is both amenable to patients as well as HCPs.

HCPs were divided regarding when the device should be used, with some suggesting patients might appreciate a calming device in the waiting room. Other HCPs suggested the treatment room, which conforms with patient views. The varying views are reflected in the literature which suggests that there is not a particular time that is anxiety-provoking for patients, and that anxiety can start a few days before the procedure and end days after.[29 30] Clarification on where exactly anxiety is most often experienced would be an ideal topic for future qualitative focus group studies.

If designed around the treatment room, HCPs suggest the device be left in situ attached to the radiotherapy couch. The easy accessibility may benefit some patients who are anxious but do not report it. For example, patients from black, Asian, and minority ethnic (BAME) communities are less likely to disclose and seek help for mental health issues but are at higher risk of mental health disorders. Reasons for not disclosing this information include not recognising or being aware of the symptoms.[31 32] The Department of Health has recognised the need to close the healthcare gap for BAME populations.[30] Leaving the device in situ would help address procedural anxiety for those who are against disclosing that they are anxious.

Patients appreciated the simulated affective touch aspect of the device. Vibration and pulsation sensations were valued for their relaxing comfort and distraction, whereas 'stroking' motions were described as too human to be performed by a machine. The idea that simulated affective touch has the potential to relieve psychological distress is supported by the literature.[16] However, it may be important not to make it too realistic as to fall into the hypothesised 'uncanny valley'[33] of imperfectly representing human features and actions.

Designing a device that is warm to touch was the idea most universally supported by both participant cohorts. Heat therapy has been shown to reduce the amount of serum cortisol and norepinephrine levels, hence downregulating the overactivated sympathetic nervous system in anxiety.[34] Moreover, studies on various populations have demonstrated that heat therapy is an effective anxiolytic intervention in medical and non-medical situations.[11 12 34]

HCPs emphasised the importance of a non-clinical visual appearance, particularly if applying the device to the arm. They were concerned patients may associate it with a blood pressure cuff, a device that can induce spike anxiety in certain patients.[35] On the other hand, the only patient requirement was to make the device look visually appealing using lights and colour. Aesthetic factors have been suggested as important considerations in the healthcare setting because they may influence whether patients will use the device.[36]

HCPs often gave their views on device design from a professional standpoint, for example, by suggesting a design which can be applied to multiple different areas of the body. This would allow personalisation of the device so patients can use the device wherever they feel might be most comforting. Radiotherapy for head and neck cancer can be particularly anxiety-inducing due to the need for a close-fitting mask,[13] and it may be prudent to make sure any device design accommodates these patients.

If this device proves to be beneficial for patients undergoing radiotherapy, it may have the potential to be used in other clinical settings to alleviate anxiety. MRI scans are known to cause anxiety,[37] and there have been investigations into interventions prior to the procedure to alleviate anxiety. Interventions have included phone calls to patients or videos for patients to watch which explain the procedure.[13] These interventions cannot be delivered in situ during the procedure,[13] which is something that an SAT device could achieve, thereby enhancing its potential efficacy.

More generally, the literature suggests that there is a need for an appropriate anxiety-alleviating device in the medical setting[11–13] but a solution has not previously been found. The case for a remote-controlled SAT device operated by a relative has applicability to the COVID-19 pandemic where imposed social distancing (maintaining a 2-metre distance from others) and many hospitals prohibiting in-person visitors lead to high levels of distress among patients and their families.

There are several limitations to this study. The use of convenience sampling may have resulted in a selection bias, leading to more homogeneous focus groups with under-representation or over-representation of particular cohorts. For example, all participants were selected from a single geographical area. Aside from sampling limitations, the collection of demographic data could have provided useful insight into the impact of ethnicity on design preferences for this device. The study also only considered adult participants; future studies may consider how the design of an SAT device might differ for children. Finally, a limitation of focus groups is that participants may not feel able or want to share their views, or that one or two people may dominate the discussion. We attempted to address these limitations by creating a relaxed environment and using experienced moderators.

A strength of this study is the use of two separate participant cohorts with different viewpoints (personal and professional). Patient opinions are vital to any intervention and meeting patient needs will enhance satisfaction and therefore improve the overall clinical outcome.[38] However, for the device to be successful, it needs to be designed in a way that meets the needs of the patient user and the requirements of minimal disruption to treatment delivery of the HCP, and as such further iterations require both patient and HCP involvement.

## CONCLUSION

In conclusion, we have identified there is a lack of data around the benefit of using tactile interventions as a tool to reduce procedural anxiety in radiotherapy patients. This study identified which design features of an anxiety-reducing tactile intervention would be valued by HCPs, patients and their carers. HCPs and patients agree on some features: a warm device which is flexible in where it can be used on the body (hence, accommodating patients no matter where they are receiving radiotherapy). Both

agreed that relatives should have some control over the device. HCPs wanted to ensure the device can easily be cleaned while patients were keen for the vibration and pulsation sensations to be used to put them at ease. Areas where no consensus was reached could be further explored as well as including greater PPI in the form of a project advisory group.

The authors intend to incorporate the desirable design features identified from this study to develop a design brief for a soft robotic device to alleviate procedural anxiety. The next stage would be to create a prototype device to test the usability and utility of this concept.

**Author affiliations**
[1]King's College London, London, UK
[2]Imperial College London, London, UK
[3]Brighton and Sussex Medical School, Brighton, UK
[4]Royal College of Art, London, UK
[5]Department of Surgery and Cancer, Imperial College London, London, UK
[6]Speech and Language Therapy, Imperial College Healthcare NHS Trust, London, UK
[7]Royal Marsden NHS Foundation Trust, London, UK
[8]Radiotherapy and Imaging, Institute of Cancer Research, London, UK

**Acknowledgements** We also acknowledge NHS funding to the NIHR Biomedical Research Centre at the Royal Marsden and the Institute of Cancer Research.

**Contributors** HH and YD contributed equally to the composition of this paper as well as the thematic analysis of the data and wish to be credited as co-lead authors if possible. AHM and HM were the senior authors on this project and provided directorial oversight. HM, AHM and CYZ developed the study design and focus group guide. CYZ, EJ and GC facilitated the focus groups alongside HM. CYZ also conducted initial data analysis and, alongside GC, provided feedback on draft manuscript versions.

**Funding** This research project was funded by the CRUK Convergence Science Centre at the Institute of Cancer Research, London, and Imperial College London (A26234). HM is funded by a National Institute for Health Research and Health Education England (HEE/NIHR), Senior Clinical Lecturer award (ICA-SCL-2018-04-ST2-002). GC is funded by the National Institute for Health Research (NIHR) Imperial Biomedical Research Centre and the NIHR Clinical Doctoral Research Fellowship Programme CDRF-2017-03-028/Integrated Clinical Academic Programme. AHM acknowledges support from the NIHR Imperial Biomedical Research Centre.

**Disclaimer** The views expressed in this publication are those of the authors and not necessarily those of the NHS or the National Institute for Health Research.

**Competing interests** None declared.

**Patient and public involvement** Patients and/or the public were involved in the design, or conduct, or reporting, or dissemination plans of this research. Refer to the Methods section for further details.

**Patient consent for publication** Not required.

**Ethics approval** This study involves human participants and was approved by Greater Manchester South Research Ethics Committee, and permission to conduct the research was granted by Health and Care Research Wales (REC reference: 19/NW/0607, protocol number: CCR5133, IRAS project ID: 266016). Participants gave informed consent to participate in the study before taking part.

**Provenance and peer review** Not commissioned; externally peer reviewed.

**Data availability statement** Data are available upon reasonable request. Data (focus group transcripts) are held at Royal Marsden NHS Foundation Trust and can be accessed on request by contacting: Helen.McNair@rmh.nhs.uk

**ORCID iD**
Hugo Hall http://orcid.org/0000-0002-5145-8412

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
