## [Reviewer comments · BMJ Open]

ARTICLE DETAILS

TITLE (PROVISIONAL)	PATIENT AND PRACTITIONER PERSPECTIVES ON THE DESIGN OF A SIMULATED AFFECTIVE TOUCH DEVICE TO REDUCE PROCEDURAL ANXIETY ASSOCIATED WITH RADIOTHERAPY: A QUALITATIVE STUDY
AUTHORS	Hall, Hugo; Dhuga, Yasmin; Yan Zheng, Caroline; Clunie, Gemma; Joyce, Elizabeth; McNair, Helen; McGregor, Alison

VERSION 1 – REVIEW

REVIEWER	Randell, Rebecca University of Bradford, Faculty of Health Studies
REVIEW RETURNED	12-Apr-2021

GENERAL COMMENTS	Thank you for the opportunity to review this paper. It is interesting and generally well written. I have several suggestions for improving the paper: Introduction Please provide a definition of simulated affective touch and give some examples of how such a device may work. Methods - Please give more detail about recruitment – which waiting rooms? Radiotherapy waiting rooms?- You state that purposive sampling was used. Purposive sampling involves selecting participants based on particular characteristics – so what were the characteristics? If it was based on people responding to a widely distributed flyer/email, it seems more like convenience sampling.- In the table, what is the category ‘other’?- Please give more information about the radiotherapy staff involved – what were their job roles?- It is stated that informed consent was obtained – was this written consent? If not, please state why.- I would like to know more about how the focus groups were conducted. Please provide the focus group topic guides as additional files and in the body of the manuscript give more information about the structure of the focus groups (e.g. a focus group would normally start with establishing ground rules and may conclude with agreeing with participants a summary of opinions; were any designs etc used to prompt discussion?) and example questions.- You note that two of the researchers were well known to the HCP cohort but that varied comments and depth of discussion suggests there was freedom for articulating thoughts. Not feeling comfortable to express thoughts can occur even if the researchers and participants don't know each other - please comment on this
--

	as a potential limitation of focus groups and what was done to address this. - How long did the focus groups last? Please state both the average and the range. Findings Please put the participant role (patient or carers or, for HCPs, their professional role) and participant number by the quotes. Smaller point: p.5 - you refer to 'both patient groups' - this should be participant groups
--	--

REVIEWER	Forbes, Erin University of Newcastle, School of Medicine and Public Health
REVIEW RETURNED	10-May-2021

GENERAL COMMENTS	General comments  1. Language could be improved. Some sections read very conversational. 2. The authors should include an explanation of what procedural anxiety is/ or a definition. Introduction  3. Please elaborate on why sedation isn't ideal. "implications for cost, risk and time" (page 4, lines 35-36) is too brief. 4. Elaborate on the rationale for using SAT. This is a fairly important part of the rationale in this study. The authors outline the problem of procedural anxiety well, but neglect to argue the effectiveness of the SAT and why it is worthwhile designing a SAT for use in this setting (there is only one sentence on page 4, lines 40-42). I believe the authors need to make their rationale much strong, outlining the existing evidence, before moving to exploring design features and considerations. 5. The link between procedural anxiety, and separation from caregivers needs to be clearer. The authors have made this a key focus of their rationale "and crucially, reduce the sense of social isolation associated with being apart from caregivers (page 4 lines 40-43)" without highlighting the importance. Why is that crucial? The first and only mention of separation from caregivers before this rationale is on line 34, "the reported sense of isolation associated with separation from caregivers or families". Please elaborate. 6. It is unclear what the purpose of the HCP's views are. In the aims the authors state "as key stakeholders in any potential device, is important to ensuring the device is put into use and does not interfere with their challenging workload" (Page 4 line 49). However, it is later described as "Healthcare professionals' views on what the patients want from the device" in the Venn diagram. Please make this clearer. 7. I also would have appreciated an argument for the inclusion of carers in this study. Carers play an enormous role in supporting a loved one through radiotherapy, however I am wondering what (if any) insight they could provide on a device to reduce anxiety in a procedure they have not experienced or witnessed themselves.
--

	METHODS Study design 8. Include ethics approval details (approval number) Participants and setting 9. Line 24 is unclear – it reads as three groups. 10. Change from “Purposive sampling was used to select both patient groups” to “Purposive sampling was used to select both participant groups” (Page 5, line 34). 11. Line 40 - Change wording from “Information sheets were then distributed to interested participants who then gave informed consent to join the study” to Information sheets were then distributed to interested patients and carers who then gave informed consent to join the study. They are still patients and carers until they provide consent. 12. Was the recruitment of HCP’s the same as for patients and carers, or were they recruited internally? Please clarify. Data Collection 13. Were the focus groups all the same size (5 people in each?), or did they vary in size? 14. Line 50 – “no new emerging data”. Do the authors mean no new emerging themes? 15. Table – In the table the participants are described as “patient had experience of radiotherapy”, “cared for someone with radiotherapy experience” and “other”, “worked as a healthcare practitioner in radiotherapy”. Who are the “other” people? It is not mentioned that participants were included who did not fit into the three categories of participant (patients, carer, HCP). RESULTS Patient views 16. Shape of Device – The quote does not represent the subtheme very well. Without any context (what question was the facilitator asking?), it is very confusing. “But for someone who is not so secure in that, something like that that you would just hold if you feel anxious. Because you don’t know where they are. You know they’ve left you. But like the ball, it would just give you a degree of confidence as someone who isn’t so secure in where they are.” 17. There are two quotes included in the ‘patient views’ results (pasted below) that sound as though people are referring to an existing device, discussing the visual appearance of the device, and tactile sensation of the device. Was there an example device provided to people? This is confusing. “Would it be possible to put lights or something inside it, so that it comes on... where the bits come out, colourful as well? “ “It’s clever isn’t it? If you get it in the right pulse, it’s just so calming. It is the ripple up and the ripple down, modulating.” Healthcare practitioner views 18. Controlling the device – The authors say there was concern that enabling the patient to control the device could introduce a
--	---

new anxiety on top of radiotherapy. I would have liked a quote to support that, to understand that thinking a little better.

19. Controlling the device – The following statement is confusing “Additionally, from a professional point of view, HCPs were concerned that if patients were given control of the device it could distract them from their workflow.” Can this be clarified? Is the concern about HCP’s being distracted from their workflow, or the patients? And if so, what ‘workflow’ do the patients have? If HCP, how would it distract them for the patient to have control? More information would be helpful.

Discussion

20. The authors say “contrastingly to HCPs, only one patient expressed a desire for a clean device and stated that they would not want to use a device shaped like a glove if someone else had used it.”. Cleanliness was only reported in the results of the patients views. If cleanliness was of greater importance to HCP’s, why is it not reported?

21. The authors mention increased pain perception as a result of high rates of anxiety (page 11, line 15), but don’t explain the mechanism behind this. Greater detail is needed.

22. Page 12, line 3, the authors say “Patients appreciated the simulated affective touch aspect of the device. Vibration and pulsation sensations were valued for their relaxing comfort and distraction, whereas ‘stroking’ motions were described as too human to be performed by a machine.” Again, was there an example device?

23. Page 12, line 43 – “one study suggested that interventions including phone calls or videos to explain the situation significantly reduces anxiety before an MRI.” Be more specific, and convincing. There have been plenty of interventions to reduce anxiety in MRI, many of which are used within the medical setting.

24. Page 12, line 51) – The authors argue that the device “could have applicability to the COVID-19 pandemic where family members are unable to visit their loved ones in the hospital”. This argument would be very confusing for someone reading this paper in ‘10 years’ time, as there is not explanation of social distancing & hospital restrictions.

Conclusion

25. The conclusions overstate the findings. The authors state “the results of this study support the idea that a tactile device has the potential to reduce procedural anxiety surrounding radiotherapy”. I disagree with this conclusion. The aim of this study was not to explore whether patients believed a SAT device would reduce their anxiety, rather what features patients would find most calming. It feels like there was a missed step here – At no point do the authors establish if patients WOULD find the device calming, before exploring the design. The authors also failed to provide a comprehensive summary of the existing evidence for such a device, and instead just stated that these devices have been “widely explored” and claimed that the evidence exists.

	26. I was surprised to not read any suggestions for devices in the conclusions. I expected some more guidance on what this device might look like (it feels quite abstract and vague reading about the design of this device). Where are the authors planning on going from here? Do the authors plan to trial some different devices and collect some quantitative data? The only recommendation for future research was to further explore the discrepancy between the views on the remote.
--	--

VERSION 1 – AUTHOR RESPONSE

Reviewer: 1

Prof. Rebecca Randell, University of Bradford

Comments to the Author:

Thank you for the opportunity to review this paper. It is interesting and generally well written. I have several suggestions for improving the paper:

Introduction

1. Please provide a definition of simulated affective touch and give some examples of how such a device may work.

More detail regarding simulated affective touch as well as a definition and examples of how such a device may work has been added to the introductory section (page 5, line 41). Please find below the relevant section from the updated manuscript:

‘Research indicates that human affective (or empathetic) touch “provides both psychological and physical comfort” Peled-Avron et al (14). However, even prior to restrictions imposed on visiting by the COVID-19 pandemic, relatives and carers have been prohibited from being in the radiotherapy room. As such, this research focuses on patient’s perspectives of a simulated affective touch (SAT) device - a haptic device that uses tactile simulations to mimic attributes of human affective touch.

Such tactile interventions have not been widely explored as a tool to reduce procedural anxiety in radiotherapy patients. However, haptic devices that simulate attributes of human affective touch, such as gentle stroking, have been shown to produce pleasant sensation (15), relieve psychological distress (16) and reduce the sense of social isolation associated with being apart from caregivers (17-19). Thus, it is postulated that SAT can provide a non-invasive intervention to combat procedural anxiety surrounding radiotherapy due to their ability to reproduce natural touch (20).’

Methods

2. Please give more detail about recruitment – which waiting rooms? Radiotherapy waiting rooms?

Page 7, Line 22: More detail around recruitment has been added. Please find below the relevant section from the updated manuscript:

“Patients and their carers were contacted using flyers placed in radiotherapy treatment and pre-treatment waiting rooms, while HCPs were internally recruited by departmental email. Patients from the local Biomedical Research Centre (BRC) Patient and Public Involvement (PPI) groups were also emailed. Information sheets were distributed to interested patients and carers who then gave written informed consent to join the study. Informed written consent was also obtained from HCPs participating in the study.”

3. You state that purposive sampling was used. Purposive sampling involves selecting participants based on particular characteristics – so what were the characteristics? If it was based on people responding to a widely distributed flyer/email, it seems more like convenience sampling.

Thank you for this comment, on reflection the authors agree that the sampling used was closer to convenience than purposive – this has been changed in the text (page 7, line 18). Please find the relevant passage from the updated manuscript below:

“Convenience sampling was used to select both participant groups.”

4. In the table, what is the category ‘other’?

Thank you. In the rubric of table 1 we have clarified what patient group ‘other’ refers to (page 17, line 8). Please find below the relevant section from the updated manuscript:

“‘Other’ refers to cancer patients recruited from the BRC Patient and public involvement group who had received imaging (e.g. MRI) but not radiotherapy.”

5. Please give more information about the radiotherapy staff involved – what were their job roles?

Further information has been provided about the job roles of the radiotherapy staff involved (page 7, line 37). The updated portion of text has been copied below for reference:

“Of the 10 HCPs, nine were therapeutic radiographers (eight involved in the planning or treatment of radiotherapy patients and one educational lead) and one was a diagnostic radiographer working mainly in magnetic resonance imaging.”

6. It is stated that informed consent was obtained – was this written consent? If not, please state why.

Page 7, line 27: Thank you for this point. Please find below the relevant section from the updated manuscript:

“Information sheets were distributed to interested patients and carers who then gave written informed consent to join the study. Informed written consent was also obtained from HCPs participating in the study.”

7. I would like to know more about how the focus groups were conducted. Please provide the focus group topic guides as additional files and in the body of the manuscript give more information about the structure of the focus groups (e.g. a focus group would normally start with establishing ground rules and may conclude with agreeing with participants a summary of opinions; were any designs etc used to prompt discussion?) and example questions.

Page 6, Line 31: Thank you for this comment. The Focus group topic guides have been attached as additional files.

We have also written another section which includes more detail about the structure of the focus groups and included some example questions. Please find below the relevant section from the updated manuscript:

“The purpose of the focus group was to establish participant thoughts around the experience of soft robotic tactile intervention probing artefacts. First, we established the ground rules and agenda of the focus group. Four probing artefacts in different shapes, touch patterns and interaction options were then introduced to the group to serve as prompts. This enabled participants to engage in discussion around the desirable and undesirable features of a tactile intervention device.

Example questions included:

- A. How participants feel about the sensations*
- B. Whether they would consider the device useful in helping to alleviate possible anxiety during the care journey*
- C. Where during the care journey they think this device would be useful*
- D. What sort of sensations they liked and will prefer”*

8. You note that two of the researchers were well known to the HCP cohort but that varied comments and depth of discussion suggests there was freedom for articulating thoughts. Not feeling comfortable to express thoughts can occur even if the researchers and participants don't know each other - please comment on this as a potential limitation of focus groups and what was done to address this.

Thank you for bringing up this important point. We have added in more detail about how the potential limitation of two of the researchers being well known to the HCP cohort was addressed (page 8, line 40). Please see the change copied from the manuscript below:

“Three qualitative researchers of mixed experience and a design researcher were responsible for facilitating the focus groups. All three qualitative researchers had no prior relationship with the patient participants but two of them were well known to the HCP cohort. To address this potential limitation, an independent researcher led all the focus groups - the researchers known to the participants only attended one of the groups. The varied comments and depth of discussion implies that that there was

sufficient freedom for articulating thoughts in each of the groups. In addition, the design researcher was not known to any participants.”

9. How long did the focus groups last? Please state both the average and the range.

Page 7, line 40: The average and range of timing for the focus groups have been added to page 6. Please find below the relevant section from the updated manuscript:

“The patient and caregiver focus groups lasted on average 83 minutes (ranging from 82 – 85 minutes). The HCP focus groups lasted on average 49 minutes (ranging from 48 – 50 minutes).”

Findings

10. Please put the participant role (patient or carers or, for HCPs, their professional role) and participant number by the quotes.

Thank you for suggesting this improvement, participant number and professional role have now been added under each of the quotations in the updated manuscript.

Smaller point:

11. p.5 - you refer to 'both patient groups' - this should be participant groups.

Page 7, Line 18: Thank you for the correction. Please find below the relevant section from the updated manuscript:

“Convenience sampling was used to select both participant groups.”

Reviewer: 2

Miss Erin Forbes, University of Newcastle

Comments to the Author:

General comments

1. Language could be improved. Some sections read very conversational.

Thank you for your comment. All authors have reviewed the manuscript and improved language where necessary. These adjustments can be seen on the track changes version of the updated

manuscript.

2. The authors should include an explanation of what procedural anxiety is/ or a definition.

Page 5, line 16: Thank you, we have added a definition of procedural anxiety. Please find below the relevant section from the updated manuscript:

“Procedural anxiety refers to excessive worry or fear of medical procedures (6).”

Introduction

3. Please elaborate on why sedation isn't ideal. “implications for cost, risk and time” (page 4, lines 35-36) is too brief.

Thank you for highlighting this deficiency, an expanded section has been added to the relevant section (page 5, line 35). The relevant section of the updated manuscript has been copied below:

‘Sedation is commonly used to tackle procedural anxiety in the context of MRI but has implications for cost, risk and time (9). This has implications in the context of daily radiotherapy treatment. On top of increased risk to the patient, sedation requires the presence of an anaesthetics team which has time implications for the treatment, facility and hospital resources.’

4. Elaborate on the rationale for using SAT. This is a fairly important part of the rationale in this study. The authors outline the problem of procedural anxiety well, but neglect to argue the effectiveness of the SAT and why it is worthwhile designing a SAT for use in this setting (there is only one sentence on page 4, lines 40-42). I believe the authors need to make their rationale much strong, outlining the existing evidence, before moving to exploring design features and considerations.

Thank you for this comment asking for elaboration on the rationale of using SAT. This has now been addressed much more fully (page 5, line 41) as well as including a definition of SAT as requested by reviewer 1. Arguments around use of SAT centre around the effectiveness of affective touch in but the inability for relatives or carers to be present in the radiotherapy room. Please find the copied section of text from the updated manuscript below for review:

‘Research indicates that human affective (or empathetic) touch “provides both psychological and physical comfort” Peled-Avron et al (14). However, even prior to restrictions imposed on visiting by the COVID-19 pandemic, relatives and carers have been prohibited from being in the radiotherapy room. As such, this research focuses on patient’s perspectives of a simulated affective touch (SAT) device - a haptic device that uses tactile simulations to mimic attributes of human affective touch.

Such tactile interventions have not been widely explored as a tool to reduce procedural anxiety in radiotherapy patients. However, haptic devices that simulate attributes of human affective touch, such as gentle stroking, have been shown to produce pleasant sensation (15), relieve psychological distress (16) and reduce the sense of social isolation associated with being apart from caregivers (17-

19). Thus, it is postulated that SAT can provide a non-invasive intervention to combat procedural anxiety surrounding radiotherapy due to their ability to reproduce natural touch (20).’

5. The link between procedural anxiety, and separation from caregivers needs to be clearer. The authors have made this a key focus of their rationale “and crucially, reduce the sense of social isolation associated with being apart from caregivers (page 4 lines 40-43)” without highlighting the importance. Why is that crucial? The first and only mention of separation from caregivers before this rationale is on line 34, “the reported sense of isolation associated with separation from caregivers or families”. Please elaborate.

Thank you for picking up on this point. In response to your comment, separation from caregivers has been de-emphasised and ‘crucially’ has been removed from page 5, line 54. Whilst the authors have noticed anecdotally that many patients ask for their loved ones to go into the radiotherapy room with them, the authors believe it only to be one part of the problem of procedural anxiety rather than a crucial element. Please see below the relevant passage from the updated manuscript:

“relieve psychological distress (16) and ~~crucially~~, reduce the sense of social isolation associated with being apart from caregivers (17-19).”

6. It is unclear what the purpose of the HCP’s views are. In the aims the authors state “as key stakeholders in any potential device, is important to ensuring the device is put into use and does not interfere with their challenging workload” (Page 4 line 49). However, it is later described as “Healthcare professionals’ views on what the patients want from the device” in the Venn diagram. Please make this clearer.

Page 6, line 12: We apologise for the confusion. We have clarified that there are multiple purposes of finding out what the HCP’s views are, both to describe the ideal features from a professional standpoint as well as using their experience to suggest features that patients might appreciate. Most importantly, since HCPs are key stakeholders, they can help translate the device into the clinical environment. Please find below the relevant section from the updated manuscript:

‘Finally, the views of HCPs as key stakeholders in any potential device were collected to ensure future clinical usability and translation. HCPs were able to advise both on professional requirements (e.g. cleanliness), as well as what they thought patients might appreciate about any calming SAT device.’

7. I also would have appreciated an argument for the inclusion of carers in this study. Carers play an enormous role in supporting a loved one through radiotherapy, however I am wondering what (if any) insight they could provide on a device to reduce anxiety in a procedure they have not experienced or witnessed themselves.

Page 6, line 6: Thank you for this comment. We have added an argument to express why we have included carers in this study. Please find below the relevant section from the updated manuscript:

‘Carers play a vital role in supporting their loved ones through difficult times, a view endorsed by a recent study (21) which reported 74% of carers supported their loved ones by talking and listening to them. Therefore, we can assume they have insight on the impact of medical procedures on their loved ones and also have a role in the requirements of such a device.’

Methods – study design

8. Include ethics approval details (approval number)

Page 6, line 57: Thank you for this comment, we have added ethics approval details. Please find below the relevant section from the updated manuscript:

“REC reference: 19/NW/0607, Protocol number: CCR5133, IRAS project ID: 266016”.

Methods - Participants and setting

9. Line 24 is unclear – it reads as three groups.

The section in question has been clarified as requested (page 7, line 8). Please see a copy of the relevant passage from the manuscript below:

“Participants were recruited from two groups of stakeholders: one comprised both patients and carers, the other group comprised health care professionals.”

10. Change from “Purposive sampling was used to select both patient groups” to “Purposive sampling was used to select both participant groups” (Page 5, line 34).

Thank you for this correction, the relevant section has been changed within the manuscript along with the change requested by reviewer 1 from purposive to convenience sampling (page 7, line 18). Please find the relevant section from the updated manuscript quoted below:

‘Convenience sampling was used to select both participant groups.’

11. Line 40 - Change wording from “Information sheets were then distributed to interested participants who then gave informed consent to join the study” to Information sheets were then distributed to interested patients and carers who then gave informed consent to join the study. They are still patients and carers until they provide consent.

Page 7, line 27: Thank you for this comment, the relevant section has been changed. Please find the relevant section from the updated manuscript quoted below:

‘Information sheets were distributed to interested patients and carers who then gave written informed consent to join the study.’

12. Was the recruitment of HCP’s the same as for patients and carers, or were they recruited internally? Please clarify.

This has been clarified in line with your comments (page 7, line 22). Please see the relevant portion of the updated manuscript below:

'Patients and their carers were contacted using flyers places in radiotherapy treatment and pre-treatment waiting rooms, while HCPs were internally recruited by departmental email.'

Methods - Data Collection

13. Were the focus groups all the same size (5 people in each?), or did they vary in size?

Information regarding focus group sizes has now been reported in the text (page 7, line 34). Please find the relevant section from the manuscript copied below:

'A total of six focus groups took place at the Royal Marsden Hospital. The total number of participants was 30, with four patient and caregiver groups (two groups with four participants, two groups with six participants) and two HCP groups (five participants each).'

14. Line 50 – “no new emerging data”. Do the authors mean no new emerging themes?

Page 7, line 44: Thank you for the correction – ‘no new emerging data’ has been changed to ‘no new emerging themes’.

15. Table – In the table the participants are described as “patient had experience of radiotherapy”, “cared for someone with radiotherapy experience” and “other”, “worked as a healthcare practitioner in radiotherapy”. Who are the “other” people? It is not mentioned that participants were included who did not fit into the three categories of participant (patients, carer, HCP).

Thank you. In the rubric of table 1 we have clarified what patient group ‘other’ refers to (page 17, line 8). Please find below the relevant section from the updated manuscript:

“‘Other’ refers to cancer patients recruited from the BRC Patient and public involvement group who had received imaging (e.g. MRI) but not radiotherapy.”

Results – patient views

16. Shape of Device – The quote does not represent the subtheme very well. Without any context (what question was the facilitator asking?), it is very confusing.

“But for someone who is not so secure in that, something like that that you would just hold if you feel anxious. Because you don’t know where they are. You know they’ve left you. But like the ball, it would just give you a degree of confidence as someone who isn’t so secure in where they are.”

Thank you for this comment. The section referenced has been changed (page 10, line 44) in line with our changes to the methods, which clarify that probing artifacts were provided to the focus group attendees so as to get their opinions on potential future design spec. The quote has also been shortened to clarify its focus on the shape of the object being a positive feature. Please find the amended section of the manuscript copied below for your reference:

“Shape of the device:

Patients expressed interest in the shape of the device. The ball shaped probing artifact received good feedback with several participants expressing positive sentiments towards it based on its shape. Other participants suggested the shape of a teddy bear or comforting soft toy to help make associations with happier times.

“the ball, it would just give you a degree of confidence as someone who isn’t so secure in where they are.”

-P19: Patient had experience of radiotherapy”

17. There are two quotes included in the ‘patient views’ results (pasted below) that sound as though people are referring to an existing device, discussing the visual appearance of the device, and tactile sensation of the device. Was there an example device provided to people? This is confusing. “Would it be possible to put lights or something inside it, so that it comes on... where the bits come out, colourful as well? “ “It’s clever isn’t it? If you get it in the right pulse, it’s just so calming. It is the ripple up and the ripple down, modulating.”

We apologise for the confusion. The aim of this study was to function as a step prior to designing the device allowing us to understand the ideal design specification to suit the needs of each stakeholder.

We first wanted to know what the ideal device should look like, so we provided participants with prompts for them to pick out the ideal features to incorporate into a device. Please find the relevant section from the manuscript copied below:

Page 6, line 32: “The purpose of the focus group was to establish participant thoughts around the experience of soft robotic tactile intervention probing artefacts. First, we established the ground rules and agenda of the focus group. Four probing artefacts in different shapes, touch patterns and interaction options were then introduced to the group to serve as prompts. This enabled participants to engage in discussion around the desirable and undesirable features of a tactile intervention device.”

Results - Healthcare practitioner views

18. Controlling the device – The authors say there was concern that enabling the patient to control the device could introduce a new anxiety on top of radiotherapy. I would have liked a quote to support

that, to understand that thinking a little better.

In line with your comments a relevant quote has been added. The quote has been included on page 12, line 9 and copied below for reference

"I just wonder if [controlling the device] introduces too many elements... They're overwhelmed already...He's relaxed, then he switches it off, then in treatment... They then panic because they want it back on. I think it introduces too many overwhelming factors."

-P28: radiographer'

19. Controlling the device – The following statement is confusing “Additionally, from a professional point of view, HCPs were concerned that if patients were given control of the device it could distract them from their workflow.” Can this be clarified? Is the concern about HCP’s being distracted from their workflow, or the patients? And if so, what ‘workflow’ do the patients have? If HCP, how would it distract them for the patient to have control? More information would be helpful.

Page 11, line 56: We have clarified that the concern is around HCP’s being distracted from their workflow due to the SAT device distracting the patient, thus not cooperating with the HCP when required. Please find the relevant section from the manuscript copied below:

'HCPs were concerned that if patients were given control of the device, it could distract the HCP from their workflow due to a lack of co-operation from the patient when required.'

Discussion

20. The authors say “contrastingly to HCPs, only one patient expressed a desire for a clean device and stated that they would not want to use a device shaped like a glove if someone else had used it.” Cleanliness was only reported in the results of the patients views. If cleanliness was of greater importance to HCP’s, why is it not reported?

Thank you for this comment. A section has been added to the HCP results section with relevant quotes to properly report on HCP desires for a device that was easily cleanable (page 12, line 30). Please find the relevant section of the updated manuscript copied below:

'Cleanliness of the device:

HCP were keen to have a device that was easily cleanable and wipeable, with one group listing it as their top feature required when considering the practicality of an SAT device for use in radiotherapy. This research was conducted pre-COVID-19 pandemic, and so this feature would likely be of even greater importance now that infection control procedures and the disinfection of all equipment post-use have been brought into such sharp focus.

"Everything would need to be wipeable."

-P27: radiographer

“it has to pass infection control”

-P25: radiographer

“Facilitator: Practicalities. Where would you fit the practicalities?”

P27: I would say that it’s cleanable being up at the top.

P28: It has to be top, yes...”

-P27, P28: both radiographers’

21. The authors mention increased pain perception as a result of high rates of anxiety (page 11, line 15), but don’t explain the mechanism behind this. Greater detail is needed.

Thank you for this point. The link between procedural anxiety and pain (particularly the mechanism of this link) is unclear at this time, and further research is required into this area. As such, the decision has been taken to remove this point from the manuscript (page 14, line 3). Please find the amended section copied from the updated manuscript below, and without mention of anxiety’s relationship to pain.

‘This may be due to the high rates of anxiety experienced during radiotherapy (4, 5).

[new paragraph]

Control of the device was a controversial topic...’

22. Page 12, line 3, the authors say “Patients appreciated the simulated affective touch aspect of the device. Vibration and pulsation sensations were valued for their relaxing comfort and distraction, whereas ‘stroking’ motions were described as too human to be performed by a machine.” Again, was there an example device?

We apologise for the confusion. The aim of this study was to function as a step prior to designing the device allowing us to understand the ideal design specification to suit the needs of each stakeholder.

We first wanted to know what the ideal device should look like, so we provided participants with prompts for them to pick out the ideal features to incorporate into a device. Please find the relevant section from the manuscript copied below:

Page 6, line 32: “The purpose of the focus group was to establish participant thoughts around the experience of soft robotic tactile intervention probing artefacts. First, we established the ground rules and agenda of the focus group. Four probing artefacts in different shapes, touch patterns and interaction options were then introduced to the group to serve as prompts. This enabled participants

to engage in discussion around the desirable and undesirable features of a tactile intervention device.”

23. Page 12, line 43 – “one study suggested that interventions including phone calls or videos to explain the situation significantly reduces anxiety before an MRI.” Be more specific, and convincing. There have been plenty of interventions to reduce anxiety in MRI, many of which are used within the medical setting.

This section has been changed to make it more specific as requested and been clarified to draw attention to the benefit of an SAT device, that it can be employed during a procedure rather than before.

Please find the amended section in page 15, line 30 of the updated transcript or copied below for your convenience.

‘MRI scans are known to cause anxiety (38), and there have been investigations into interventions prior to the procedure to alleviate anxiety. Interventions have included calls to or videos for patients to watch which explain the procedure (13). Whilst reducing anxiety levels, these interventions cannot be delivered in-situ during the procedure (13), which is something that an SAT device could achieve.’

24. Page 12, line 51) – The authors argue that the device “could have applicability to the COVID-19 pandemic where family members are unable to visit their loved ones in the hospital”. This argument would be very confusing for someone reading this paper in 10 years’ time, as there is not explanation of social distancing & hospital restrictions.

This has been amended in the text to give more detail as regards to the measures in place to combat the pandemic (page 15, line 40). The relevant section of the updated manuscript has been copied below for reference:

‘The case for a remote-controlled SAT device operated by a relative, has applicability to the COVID-19 pandemic where imposed social distancing (maintaining a 2-meter distance from others) and many hospitals prohibiting in-person visitors lead to high levels of distress amongst patients and their families.’

Conclusion

25. The conclusions overstate the findings. The authors state “the results of this study support the idea that a tactile device has the potential to reduce procedural anxiety surrounding radiotherapy”. I disagree with this conclusion. The aim of this study was not to explore whether patients believed a SAT device would reduce their anxiety, rather what features patients would find most calming. It feels like there was a missed step here – At no point do the authors establish if patients WOULD find the device calming, before exploring the design. The authors also failed to provide a comprehensive summary of the existing evidence for such a device, and instead just stated that these devices have been “widely explored” and claimed that the evidence exists.

Regarding the reviewer's first point, 'The conclusions overstate the findings.'

- Thank you for this comment. This paper is intended as a step prior to designing the device to enable us to find the most favourable features for us to take forward to build an ideal calming device.
- In accordance with the above, the sentence 'the results of this study support the idea that a tactile device has the potential to reduce procedural anxiety surrounding radiotherapy' has been removed.

Regarding the reviewers second point, 'there was a missed step here – at no point do the authors establish if patients would find the device calming, before exploring the design.'

- We apologise for the confusion. We haven't designed the final device yet, we were establishing what features participants would find most calming to enable us to take these design features forward to build the ideal device. Question B in the focus group guide asked specifically this question: 'Whether they would consider the device useful in helping to alleviate possible anxiety during the care journey'. While this isn't addressing the aim of the study, we asked this question to collect feedback. The response to this was positive, lending credence to our aim of learning about the features that a potential calming device design could have.

Regarding the reviewer's third point, 'The authors also failed to provide a comprehensive summary of the existing evidence for such a device, and instead just stated that these devices have been "widely explored" and claimed that the evidence exists.'

- Thank you for this comment. This has now been addressed much more fully (page 5, line 41) as well as including a definition of SAT as requested by reviewer 1. Arguments around use of SAT centre around the effectiveness of affective touch combined with the inability for relatives or carers to be present in the radiotherapy room. Please find the copied section of text from the updated manuscript below for review:

'Research indicates that human affective (or empathetic) touch "provides both psychological and physical comfort" Peled-Avron et al (14). However, even prior to restrictions imposed on visiting by the COVID-19 pandemic, relatives and carers have been prohibited from being in the radiotherapy room. As such, this research focuses on patient's perspectives of a simulated affective touch (SAT) device - a haptic device that uses tactile simulations to mimic attributes of human affective touch.

Such tactile interventions have not been widely explored as a tool to reduce procedural anxiety in radiotherapy patients. However, haptic devices that simulate attributes of human affective touch, such as gentle stroking, have been shown to produce pleasant sensation (15), relieve psychological distress (16) and reduce the sense of social isolation associated with being apart from caregivers (17-19). Thus, it is postulated that SAT can provide a non-invasive intervention to combat procedural anxiety surrounding radiotherapy due to their ability to reproduce natural touch (20).'

26. I was surprised to not read any suggestions for devices in the conclusions. I expected some more guidance on what this device might look like (it feels quite abstract and vague reading about the design of this device). Where are the authors planning on going from here? Do the authors plan to

trial some different devices and collect some quantitative data? The only recommendation for future research was to further explore the discrepancy between the views on the remote.

With regards to, “I was surprised to not read any suggestions for devices in the conclusions. I expected some more guidance on what this device might look like (it feels quite abstract and vague reading about the design of this device). “

- Thank you for this comment. Some more suggestions have been added to the conclusion based on both the patient and HCP responses to soft robotic artifacts (page 16, line 18). The relevant new sentences from the updated manuscript have been copied below:

‘Both agreed that relatives should have some control over the device. HCPs wanted to ensure the device can easily be cleaned while patients were keen for the vibration and pulsation sensations to be used to put them at ease.’

With regards to, “Where are the authors planning on going from here? Do the authors plan to trial some different devices and collect some quantitative data? The only recommendation for future research was to further explore the discrepancy between the views on the remote.”

- Thank you for this comment. We have added a section on our intentions for future work page 16, line 25. The relevant section of the updated manuscript is copied below for your convenience:

‘The authors intend to incorporate the desirable design features identified from this study to develop a design brief for a soft robotic device to alleviate procedural anxiety. The next stage would be to create a prototype device to test the usability and utility of this concept.’

Reviewer: 1

Competing interests of Reviewer: I have no competing interests to declare.

Reviewer: 2

Competing interests of Reviewer: I have no competing interests.

VERSION 2 – REVIEW

REVIEWER	Randell, Rebecca University of Bradford, Faculty of Health Studies
REVIEW RETURNED	18-Oct-2021

GENERAL COMMENTS	Thank you for addressing my comments. I only have one further comment: having acknowledged that convenience sampling was used, please discuss the limitations of this in the discussion section.
--

REVIEWER	Forbes, Erin University of Newcastle, School of Medicine and Public Health
REVIEW RETURNED	20-Oct-2021

GENERAL COMMENTS	Overall comments
------------------

	1. I think the two separate 'stakeholder' groups are still a little unclear. Sometimes the carers are mentioned, and sometimes they are not. Introduction 2. The fourth paragraph could use some restructuring. The current wording is confusing. Something along the lines of: Relatives and carers are prohibited from being in the radiotherapy treatment room, leaving patients alone. Since the COVID-19 pandemic, patients are even more isolated with many hospitals not allowing support people into waiting area. 3. The next paragraph (5) could also benefit from restructure to improve flow. If the first sentence ("such tactile interventions...") and the second sentence (However, haptic devices...") were switched positions, it would improve clarity. Haptic devices that simulate attributes of human affective touch, such as gentle stroking, have been shown to produce pleasant sensation (15), relieve psychological distress (16) and reduce the sense of social isolation associated with being apart from caregivers (17-19). However, such tactile interventions have not been widely explored as a tool to reduce procedural anxiety in radiotherapy patients. It is postulated that SAT can provide a non-invasive intervention to combat procedural anxiety surrounding radiotherapy due to their ability to reproduce natural touch (20). 4. Regarding the aims paragraph on page 6, the last two sentences are results, and do not belong in the introduction. Study design 5. The first sentence of the second paragraph is confusing. It is the first mention of 'probing artefacts' and does not read well. The authors provide an explanation below on what probing artefacts are, but it is very confusing in the initial sentence. The purpose of the focus group was to establish participant thoughts around the experience of soft robotic tactile intervention probing artefacts. Participants and setting 6. A minor comment – be consistent with the wording of 'written informed consent' and 'informed written consent', especially when they are presented near each other in the paragraph. It is clunky to read. Data collection 7. The explanation of the focus groups is confusing. The authors explain that three qualitative researchers facilitate the focus groups, and then describe that and independent researchers led all of the focus groups? Please clarify the difference in the roles. Patient and public involvement 8. NHS PPI groups – what is this? RESULTS 9. The authors describe that the data identified SIX key sub themes, however in the PATIENT VIEWS section there are SEVEN key sub themes outlined. Tactile sensation of the device appears to be omitted in the initial listing of the subthemes. 10. List the seven key subthemes based on what HCPs believe patients would like in the same order they are presented in the text.
--	---

	PATIENT VIEWS 11. Several of the key subthemes begin with statements about how the patients views differ from the HCPs. As the HCP results have not been presented to the reader at this point, this should be removed. If the authors wish to point of the discrepancies, include these statements in the HCP results (e.g. contrary to what patients reported as important, HCP's believed patients etc etc) HEALTHCARE PRACTITIONER VIEWS 12. In the control of the device section, the last sentence before the quotes is clunky and confusing (line 58). DISCUSSION 13. The argument about 'current interventions in place' could use some work. The research is limited, not the number of interventions. We don't necessarily need a plethora of available interventions. I would also argue there are no 'interventions in place'. This reads as though there are interventions that are universally used in RT departments. This isn't the case. 14. The first sentence in the second paragraph (line 45) is a little presumptuous. We don't yet know that the device will alleviate anxiety – perhaps reword to say that it 'may' alleviate anxiety. 15. In the third paragraph, the authors state “a phenomenon commented upon by Crowe et al”. What does this tell us? It is confusing and uninformative. 16. In the fourth paragraph the authors conclude that patients and their carers focused on anxiety relieving features – “this may be due to the high rates of anxiety experiences during radiotherapy”. This seems a strange conclusion to draw when this was the purpose of the study. 17. The sixth paragraph doesn't flow well. The first half of the paragraph (regarding the control of the device) is quite specific and relates to the previous paragraph, and the second half of the paragraph (future research) seems like a sweeping statement of the whole study. 18. Page 15, line 20-21 – this is incorrect. The device would not be used wherever on the body the patient is having radiotherapy. They would be avoiding the area on the body the patient is having radiotherapy. 19. Page 15, line 29. The authors state “both cohorts had various suggestions about other occasions the device could be used...”. This wasn't introduced in the results, so don't refer introduce it in the discussion to make an argument.
--	--

VERSION 2 – AUTHOR RESPONSE

Reviewer: 1
Prof. Rebecca Randell, University of Bradford

Comments to the Author:

Thank you for addressing my comments. I only have one further comment: having acknowledged that convenience sampling was used, please discuss the limitations of this in the discussion section.

Thank you for your further comment. A section discussing convenience sampling has been added to the 'limitations' section in the discussion. Please find the relevant section copied below for your convenience:

(Page 15, Line 36): 'The use of convenience sampling may have resulted in a selection bias, leading to more homogenous focus groups with under- or over-representation of particular cohorts. For example, all participants were selected from a single geographical area.'

Reviewer: 2
Miss Erin Forbes, University of Newcastle

Comments to the Author:
Overall comments

1. I think the two separate 'stakeholder' groups are still a little unclear. Sometimes the carers are mentioned, and sometimes they are not.

Thank you for your comment. The two separate stakeholder groups are as follows:

1. Patients and carers
2. Healthcare practitioners

We have clarified this throughout the text as required. The most obvious of these clarifications is in the renaming of the 1st results section to 'Patient and Carer Views' rather than simply 'Patient Views'. All changes are detailed in the tracked changes version of the manuscript.

Introduction

2. The fourth paragraph could use some restructuring. The current wording is confusing. Something along the lines of:

Relatives and carers are prohibited from being in the radiotherapy treatment room, leaving patients alone. Since the COVID-19 pandemic, patients are even more isolated with many hospitals not allowing support people into waiting areas.

Thank you for your comment, the section has been changed to your suggested wording.

(Page 5, line 43): 'However, relatives and carers are prohibited from being in the radiotherapy treatment room, leaving patients alone. Since the COVID-19 pandemic, patients are even more isolated with many hospitals not allowing visitors into waiting areas.'

3. The next paragraph (5) could also benefit from restructure to improve flow. If the first sentence ("such tactile interventions...") and the second sentence (However, haptic devices...") were switched positions, it would improve clarity.

Thank you for your comment, the section has been changed as per your suggestion and the first and second sentences switched.

(Page 5, line 51). 'Haptic devices simulate attributes of human affective touch, such as gentle stroking, which has been shown to produce pleasant sensation (15), relieve psychological distress (16) and reduce the sense of social isolation associated with being apart from caregivers (17-19).'

However, such tactile interventions have not been widely explored as a tool to reduce procedural anxiety in radiotherapy patients. It is postulated that SAT can provide a non- invasive intervention to combat procedural anxiety surrounding radiotherapy due to their ability to reproduce natural touch (20).'

4. Regarding the aims paragraph on page 6, the last two sentences are results, and do not belong in the introduction.

Thank you for your comment. To explain the rationale of these two sentences and why the authors believe they belong in the introduction:

1. The first of these sentences is intended as an explanation of why HCP views are key to the design of any SAT device
2. The second is intended to explain our further plans with regards to utilising the data collected.

We have changed the sentences in question in order to clarify this (Page 6, line 13).

Please find the updated section copied verbatim below:

(Page 6, line 13): 'HCP views were collected in order to advise on professional requirements (e.g. cleanability), as well as what they thought patients might appreciate about any calming SAT device. We intend to use the insights of the above groups to facilitate the future development of a design brief for an SAT prototype device to reduce procedural anxiety. This will ensure that the prototype device will be relevant and suitable for future clinical settings.'

Study design

5. The first sentence of the second paragraph is confusing. It is the first mention of 'probing artefacts' and does not read well. The authors provide an explanation below on what probing artefacts are, but it is very confusing in the initial sentence.

Thank you for your comment. We have revised these sentences to increase clarity. The sentences now read:

(Page 6, line 33): 'The purpose of the focus group was to establish participant thoughts around the experience of four soft robotic tactile intervention probing artefacts, which all had different shapes, touch patterns and interaction options. First, we established the ground rules and agenda of the focus group. Then the four probing artefacts were introduced to the group to serve as prompts.'

Participants and setting

6. A minor comment – be consistent with the wording of 'written informed consent' and 'informed written consent', especially when they are presented near each other in the paragraph. It is clunky to read.

Thank you for your comment, we have altered this section so that it now reads:

(Page 7, line 24): 'Information sheets were distributed to interested patients and carers who then gave written informed consent to join the study. Written informed consent was also obtained from HCPs participating in the study.'

Data collection

7. The explanation of the focus groups is confusing. The authors explain that three qualitative researchers facilitate the focus groups, and then describe that and independent researchers led all of the focus groups? Please clarify the difference in the roles.

Apologies for the confusion and thank you for drawing this to our attention. We have clarified and simplified the section detailing how the focus groups were run (Page 8, line 36). We have also moved the section regarding limitations of the focus groups, to the limitations section of the discussion (Page 15, line 44).

The relevant passages have been copied below verbatim for your convenience.

Data collection (Page 8, line 36):

'Each focus group was conducted by two facilitators with qualitative cancer research experience and were supported by the design researcher. All qualitative researchers involved in leading the focus groups had no prior relationship with the patient participants but two of them were well known to the HCP cohort. The varied comments and depth of discussion implies that that there was sufficient freedom for articulating thoughts in each of the groups. The design researcher was not known to any participants.'

Limitations (Page 15, line 44):

'A limitation of focus groups is that participants may not feel able or want to share their views or that one or two people may dominate the discussion. We attempted to address these limitations by creating a relaxed environment and using experienced moderators.'

Patient and public involvement

8. NHS PPI groups – what is this?

In this context 'PPI' refers to Patient and Public Involvement groups, the authors felt it was key to involve them in the study planning. PPI was defined on (Page 7, line 24) in reference to Biomedical Research Centre (BRC) Patient and Public Involvement (PPI) groups, however we have redefined it with relation to the NHS PPI groups in order to improve clarity (Page 9, line 7).

RESULTS

9. The authors describe that the data identified SIX key sub themes, however in the PATIENT VIEWS section there are SEVEN key sub themes outlined. Tactile sensation of the device appears to be omitted in the initial listing of the subthemes.

Thank you for commenting on this error: We have revised the sentence, so it now reads:

(Page 9, line 19): *'Further analysis of the data identified seven key sub-themes based on what patients wanted from the device; control of the device, temperature of the device, cleanliness of the device, where the device should be located on the body, shape of the device, and visual appearance of the device and tactile sensation of the device.'*

10. List the seven key subthemes based on what HCPs believe patients would like in the same order they are presented in the text.

Thank you for your comment. We have rewritten the paragraph so that the sub themes appear in the same order as presented in the text. The sentence now reads:

(Page 9, line 30): *‘Finally, there were eight key sub-themes based on what HCPs believe patients would like; non-clinical appearance of the device, fitting the device into an object found in the treatment room, who controls the device, where the device should be located on the body, cleanliness of the device, temperature of the device, when to use the device and customisation of the device. These themes are summarised in figure 1 and discussed in more detail below.’*

PATIENT VIEWS

11. Several of the key subthemes begin with statements about how the patients views differ from the HCPs. As the HCP results have not been presented to the reader at this point, this should be removed. If the authors wish to point of the discrepancies, include these statements in the HCP results (e.g. contrary to what patients reported as important, HCP’s believed patients etc etc)

Thank you for your suggestion. We have revised the ‘Patient and Carer Results’ section in line with your comment and removed these comparative statements. Where appropriate, we have added the comparative statements to the HCP section instead. An example of this would be the point regarding warmth – the comparison between groups is now drawn in the HCP results section:

(Page 12, line 50): *‘In a point of similarity to the patients and carers, most HCPs believed that a warm device would be calming to a patient. However, a few HCPs disagreed and suggested that it would be good to adjust the temperature based on the needs of the patient.’*

HEALTHCARE PRACTITIONER VIEWS

12. In the control of the device section, the last sentence before the quotes is clunky and confusing (line 58).

Thank you for suggesting this improvement. We have altered the sentence to increase clarity. The sentence now reads:

(Page 11, line 50): *‘Additionally, from a professional point of view, HCPs were concerned about giving patients control of the device. They were worried that the device could distract patients and affect their ability to remain still for the radiotherapy treatment or imaging.’*

DISCUSSION

13. The argument about ‘current interventions in place’ could use some work. The research is limited, not the number of interventions. We don’t necessarily need a plethora of available interventions. I would also argue there are no ‘interventions in place’. This reads as though there are interventions that are universally used in RT departments. This isn’t the case.

Thank you for your comment. We have altered the sentence to increase clarity. The updated portion of the manuscript is included verbatim below:

(Page 13, line 33): *'On the non-pharmacological side, there are no universally used interventions in radiotherapy departments, and research into new interventions is limited (25). Further, any non-pharmacological interventions currently being investigated do not make up for the lack of physical interaction experienced by patients, instead focusing on distracting patients (for example through music).'*

14. The first sentence in the second paragraph (line 45) is a little presumptuous. We don't yet know that the device will alleviate anxiety – perhaps reword to say that it 'may' alleviate anxiety.

Thank you for your comment. We have altered the sentence to suggest the device may relieve anxiety. The sentence now reads:

(Page 13, line 40): *'We have identified key design aspects for an SAT device that may alleviate the anxiety patients experience during radiotherapy and/or imaging.'*

15. In the third paragraph, the authors state "a phenomenon commented upon by Crowe et al". What does this tell us? It is confusing and uninformative.

Thank you for your comment. We have altered this sentence to increase clarity. The sentence now reads:

(Page 13, line 47): *'As suggested by Crowe et al. (26), this could have been because of their different roles (HCP versus patient/carer) and, therefore, different priorities.'*

16. In the fourth paragraph the authors conclude that patients and their carers focused on anxiety relieving features – "this may be due to the high rates of anxiety experiences during radiotherapy". This seems a strange conclusion to draw when this was the purpose of the study.

Thank you for your comment. We have removed the relevant section, the fourth paragraph in its entirety is copied below for your convenience:

(Page 13, line 51): *'Perhaps unsurprisingly, HCPs were concerned about the device not disrupting their workflow. For example, they emphasised designing an easily cleanable device, this is consistent with the literature which suggests interruptions to their workload may contribute to medical errors (27).'*

17. The sixth paragraph doesn't flow well. The first half of the paragraph (regarding the control of the device) is quite specific and relates to the previous paragraph, and the second half of the paragraph (future research) seems like a sweeping statement of the whole study.

1 - The first half of the paragraph (regarding the control of the device) is quite specific and relates to the previous paragraph.

Thank you for your comment. On consideration we have added the part of the 6th paragraph regarding control of the device to the previous paragraph.

2- The second half of the paragraph (future research) seems like a sweeping statement of the whole study.

The wording has been reconsidered and now reads:

"Future research should incorporate further patient and public involvement and engagement, perhaps in the form of a project advisory group"

Both paragraph 5 and 6 are included below in their revised form:

(Page 13, line 57): *‘Control of the device was a controversial topic. Patients expressed the desire to have control of the device, whereas HCPs were keen for patients not to have control. HCP feared that patient control would lead to disruptions to the treatment session. Both groups agreed that allowing relatives some degree of control of the device would be good. This finding is supported by other studies that have shown the involvement of friends and family in radiotherapy treatment has a positive impact on patient care (28). This controversy on control of the device warrants further exploration in future research. Specifically, how useful a device to reduce anxiety would be if it was controlled by the HCP against the patient’s preference – relative control may provide a happy middle ground in this regard.*

Future research should also incorporate further patient and public involvement and engagement, perhaps in the form of a project advisory group. Involving such an advisory group at all stages of the research could help to make sure the patient perspective is not neglected (29), ensuring any device is both amenable to patients as well as HCPs.’

18. Page 15, line 20-21 – this is incorrect. The device would not be used wherever on the body the patient is having radiotherapy. They would be avoiding the area on the body the patient is having radiotherapy.

Thank you for your comment. We appreciate the confusion inherent in the previous wording and have altered the sentence to better reflect our intention:

(Page 15, line 12): *‘This would allow personalisation of the device so patients can use the device wherever they feel most comforting.’*

19. Page 15, line 29. The authors state “both cohorts had various suggestions about other occasions the device could be used...”. This wasn’t introduced in the results, so don’t refer introduce it in the discussion to make an argument.

Thank you for your comment. We have removed the relevant sentence (Page 15, line 18).

Reviewer: 1

Competing interests of Reviewer: I have no competing interests to declare

Reviewer: 2

Competing interests of Reviewer: No competing interests

VERSION 3 – REVIEW

REVIEWER	Forbes, Erin University of Newcastle, School of Medicine and Public Health
REVIEW RETURNED	20-Dec-2021

GENERAL COMMENTS	This manuscript has improved significantly. However, there are still areas that could be improved:
--

ABSTRACT

Page 2, line 29

“Different priorities of patients, carers and healthcare practitioners were evident”

This isn't a correct statement – the results were analysed in patient/carer perspectives and healthcare (i.e. the 'different priorities of carers from patients were not evident). Refer to the groups consistently.

INTRODUCTION

Page 4, line 23

This paragraph has an odd structure. The first sentence does not seem to fit with the rest of the paragraph. I also wonder if this sentence should be reworded *“However, the consequences of procedural anxiety extend beyond radiotherapy and include diagnostic imaging, and other invasive procedures performed on the conscious patient”*.

Do you mean the experience of procedural anxiety extends beyond RT, instead of consequences?

Page 4, line 35

“Sedation is commonly used to tackle procedural anxiety in the context of MRI but has implications for cost, risk and time (9). This has implications in the context of daily radiotherapy treatment. On top of increased risk to the patient, sedation requires the presence of an anaesthetics team which has time implications for the treatment, facility and hospital resources.”

Consider rewording. The use of the word 'implications' is repetitive. Also, the authors refer to the increased risk to the patient as though it has already been discussed, but there is an explanation of what the increase risk is?

Page 7 line 3,

The column for 'Other' has an asterisk next to it. Is there an explanation of what 'other' means somewhere in a footnote?

Page 7, line 54

Are the researchers 'qualitative researchers' or 'medical researchers'? Be consistent.

RESULTS

Page 8, line 25

The authors report *"two key sub-themes were identified based on what HCPs would want from the device from a professional point of view: functionality of the device and universal compatibility of the device."*

However, these two sub themes don't seem to be reported, only the eight key subthemes on what the HCPs believe patients would want?

PATIENT AND CARER VIEWS

Control of the device:

Page 8, line 47 *"Patients and carers reached strong consensus in favour of being able to control the device themselves."*

Do you mean that the patient should be able to control the device? If carers are speaking on behalf of patients, perhaps change to *"Patients and carers reached strong consensus in favour of the patient being able to control the device themselves. They highlighted that in times of uncertainty, the device would give the patient something to be in control of."*

Shape of the device:

Page 9, line 41

Typographical error 'Artifact'

Cleanliness of the device:

Page 11, line 26

"In contrast to patients, HCP were keen to have a device that was easily cleanable..."

Although it was only one carer in the patients and carer group who identified cleanliness, the authors have reported it as a 'key subtheme'. It seems odd to the report that it is a contrast to patients that HCP are keen to have a device that is easily

cleanable. I would only report 'in contrast' if the patients said they wanted it to be hard to clean.

DISCUSSION

Page 12, line 26

Change from 'stress-provoking' to 'anxiety-provoking'. Stress and anxiety aren't the same, and there has been no mention of stress until now.

Page 12, line 35

"Further, any non-pharmacological interventions currently being investigated do not

make up for the lack of physical interaction experienced by patients, instead focusing on distracting patients (for example through music)."

Please reword - "make up for" is probably not the right wording. Perhaps 'address the lack of' might be a better phrasing.

Page 12, line 46

"Overall, we found that participant cohorts had mixed opinions on the design aspects of such a device. As suggested by Crowe et al. (26), this could have been because of their different roles (HCP versus patient/carer) and, therefore, different priorities"

This seems like an odd conclusion – wasn't the purpose of including patients, carers, and HCP because their different roles would mean different priorities?

Page 12, line 51

"Perhaps unsurprisingly, HCPs were concerned about the device not disrupting their

workflow. For example, they emphasised designing an easily cleanable device, this is consistent with the literature which suggests interruptions to their workload may contribute to medical errors (27)."

Clunky. Suggestion for rewording:

Perhaps unsurprisingly, HCPs were concerned about the device not disrupting their

workflow, for example, emphasis was placed on designing an easily cleanable device. This is consistent with the literature which suggests interruptions to their workload may contribute to medical errors (27).

Page 13, line 3

“This finding is supported by other studies that have shown the involvement of friends and family in radiotherapy treatment has a positive impact on patient care (28).”

This statement seems abit of a stretch. The family/friends involvement referred to in the Mackenzie paper is regarding better services, information and support for family and friends. I don't think this supports the point you are making.

Page 13, line 6

“This controversy on control of the device warrants further exploration in future research. Specifically, how useful a device to reduce anxiety would be if it was controlled by the HCP against the patient's preference – relative control may provide a happy middle ground in this regard.”

I think this point needs some work. The authors initially framed the subtheme of control of the device as “what the HCPs though patients would want”. Patients are reporting they want one thing, and the HCP are reporting that they think the patients would want another, shouldn't the patient preference supersede what the HCP think the patient would want?

Page 13, line 30

Add 'For example,', before *“patients from Black, Asian...”*.

Page 13, line 34

“in denial of mental health issues”

Please reword this. Procedural anxiety can be very situational, not necessarily considered a 'mental health issue'

Page 14, line 22

	Interventions have included calls to or videos for patients to watch which explain the procedure (13). Calls to who? Please specify. Page 14, line 23 “Whilst reducing anxiety levels, these interventions cannot be delivered in-situ during the procedure (13)” This sentence could be reworded. “Whilst reducing anxiety levels” implies the interventions were effective? Is this correct? It’s best to spell it out for the reader. Page 14, line 50 “The fact that this study uses two separate participant cohorts with different viewpoints (personal and professional) is a strength.” ‘The fact that’ is not a smooth introduction to a paragraph. Consider rewording to something like: A strength of this study is the use of two separate participant cohorts with different viewpoints (personal and professional). Page 14, line 53 “However, patient desires need to be balanced with HCP requirements as if the device was designed in a way that disrupted the treatment it would be unlikely to be implemented.” Please consider rewording to improve flow. Page 14, line 56 Further qualitative research could be conducted on this topic with the use of a patient and public involvement project advisory group at all stages of the study to make sure the interests of both groups are equally represented (29, 40). Repetition – consumer involvement has already been mentioned earlier in the discussion.
--	---

VERSION 3 – AUTHOR RESPONSE

We appreciate the apology regarding the delay in reaching a decision and the comment that the manuscript has improved considerably. We have responded to the additional comments as below.

ABSTRACT

Page 2, line 29

“Different priorities of patients, carers and healthcare practitioners were evident” This isn't a correct statement – the results were analysed in patient/carer perspectives and healthcare (i.e. the 'different priorities of carers from patients were not evident). Refer to the groups consistently.

We have amended as:

‘Different priorities of patients and their carers, and healthcare practitioners were evident.’

INTRODUCTION

Page 4, line 23

This paragraph has an odd structure. The first sentence does not seem to fit with the rest of the paragraph. I also wonder if this sentence should be reworded *“However, the consequences of procedural anxiety extend beyond radiotherapy and include diagnostic imaging, and other invasive procedures performed on the conscious patient”*.

Do you mean the experience of procedural anxiety extends beyond RT, instead of consequences?

We have clarified as below:

‘Procedural anxiety is not limited to radiotherapy and also occurs during diagnostic imaging, and other invasive procedures performed on a conscious patient.’

Page 4, line 35

“Sedation is commonly used to tackle procedural anxiety in the context of MRI but has implications for cost, risk and time (9). This has implications in the context of daily radiotherapy treatment. On top of increased risk to the patient, sedation requires the presence of an anaesthetics team which has time implications for the treatment, facility and hospital resources.”

Consider rewording. The use of the word 'implications' is repetitive. Also, the authors refer to the increased risk to the patient as though it has already been discussed, but there is an explanation of what the increase risk is?

We have reworded as below:

‘Sedation especially if treatment is required on a daily basis has implications on service provision, facility usage and hospital resources, and repeated sedation may have associated medical risks in some patient groups.’

Page 7 line 3,

The column for 'Other' has an asterisk next to it. Is there an explanation of what 'other' means somewhere in a footnote?

The explanation of what 'other' refers to is included in the table legend which was located at the end of the manuscript as requested by the BMJ Open submission guidance. We have moved the legend to under the table as requested by the editorial assistant. Copied below is the explanation for 'other':

**'Other' refers to cancer patients recruited from the BRC Patient and public involvement group who had received imaging (e.g. MRI) but not radiotherapy.*

Page 7, line 54

Are the researchers 'qualitative researchers' or 'medical researchers'? Be consistent.

We have clarified, the researchers are medical researchers with qualitative experience.

RESULTS

Page 8, line 25

The authors report *"two key sub-themes were identified based on what HCPs would want from the device from a professional point of view: functionality of the device and universal compatibility of the device."*

However, these two sub themes don't seem to be reported, only the eight key subthemes on what the HCPs believe patients would want?

Thank you, this was a formatting error and we have deleted the paragraph concerned.

PATIENT AND CARER VIEWS

Control of the device:

Page 8, line 47 *"Patients and carers reached strong consensus in favour of being able to control the device themselves."*

Do you mean that the patient should be able to control the device? If carers are speaking on behalf of patients, perhaps change to "Patients and carers reached strong consensus in favour of the patient being able to control the device themselves. They highlighted that in times of uncertainty, the device would give the patient something to be in control of."

Thank you for the suggestion we have amended as recommended.

Shape of the device:

Page 9, line 41 Typographical error 'Artifact'

Thank you for pointing this out, we have amended to the correct spelling.

Cleanliness of the device:

Page 11, line 26

"In contrast to patients, HCP were keen to have a device that was easily cleanable..."

Although it was only one carer in the patients and carer group who identified cleanliness, the authors have reported it as a 'key subtheme'. It seems odd to the report that it is a contrast to patients that HCP are keen to have a device that is easily cleanable. I would only report 'in contrast' if the patients said they wanted it to be hard to clean.

We have deleted 'in contrast to patients' from the relevant section in the text.

DISCUSSION

Page 12, line 26

Change from 'stress-provoking' to 'anxiety-provoking'. Stress and anxiety aren't the same, and there has been no mention of stress until now.

We have amended as suggested.

Page 12, line 35

"Further, any non-pharmacological interventions currently being investigated do not make up for the lack of physical interaction experienced by patients, instead focusing on distracting patients (for example through music)."

Please reword - "make up for" is probably not the right wording. Perhaps 'address the lack of' might be a better phrasing.

We have amended as suggested.

Page 12, line 46

"Overall, we found that participant cohorts had mixed opinions on the design aspects of such a device. As suggested by Crowe et al. (26), this could have been because of their different roles (HCP versus patient/carer) and, therefore, different priorities" This seems like an odd conclusion – wasn't the purpose of including patients, carers, and HCP because their different roles would mean different priorities?

We have added the sentence below:

'This is not unexpected as HCP have different roles and priorities to the patient and their carer.'

Page 12, line 51

"Perhaps unsurprisingly, HCPs were concerned about the device not disrupting their workflow. For example, they emphasised designing an easily cleanable device, this is consistent with the literature which suggests interruptions to their workload may contribute to medical errors (27)."

Clunky. Suggestion for rewording:

Perhaps unsurprisingly, HCPs were concerned about the device not disrupting their workflow, for example, emphasis was placed on designing an easily cleanable device. This is consistent with the

literature which suggests interruptions to their workload may contribute to medical errors (27).

We have amended as suggested.

Page 13, line 3

“This finding is supported by other studies that have shown the involvement of friends and family in radiotherapy treatment has a positive impact on patient care (28).”

This statement seems abit of a stretch. The family/friends involvement referred to in the Mackenzie paper is regarding better services, information and support for family and friends. I don't think this supports the point you are making.

We have deleted this sentence.

Page 13, line 6

“This controversy on control of the device warrants further exploration in future research. Specifically, how useful a device to reduce anxiety would be if it was controlled by the HCP against the patient's preference – relative control may provide a happy middle ground in this regard.”

I think this point needs some work. The authors initially framed the subtheme of control of the device as “what the HCPs though patients would want”. Patients are reporting they want one thing, and the HCP are reporting that they think the patients would want another, shouldn't the patient preference supersede what the HCP think the patient would want?

We have amended as below to clarify:

‘Future iterations of the device need to provide the level of control required by the patients whilst addressing HCPs concerns in relation to service delivery. If the efficacy of the device is established this may facilitate greater acceptance by the HCPs.’

Page 13, line 30

Add ‘For example,’ before *“patients from Black, Asian...”*.

We have amended as suggested.

Page 13, line 34

“in denial of mental health issues”

Please reword this. Procedural anxiety can be very situational, not necessarily considered a ‘mental health issue’

We have amended as below:

‘Reasons for not disclosing this information include not recognising or being aware of the symptoms (31, 32).’

Page 14, line 22

Interventions have included calls to or videos for patients to watch which explain the procedure (13).

Calls to who? Please specify.

We have amended as below:

'Interventions have included phone calls to patients or videos for patients to watch which explain the procedure (13).'

Page 14, line 23

"Whilst reducing anxiety levels, these interventions cannot be delivered in-situ during the procedure (13)"

This sentence could be reworded. "Whilst reducing anxiety levels" implies the interventions were effective? Is this correct? It's best to spell it out for the reader.

We have amended as below:

'These interventions cannot be delivered in-situ during the procedure (13), which is something that an SAT device could achieve, thereby enhancing its potential efficacy.'

Page 14, line 50

"The fact that this study uses two separate participant cohorts with different viewpoints (personal and professional) is a strength."

'The fact that' is not a smooth introduction to a paragraph. Consider rewording to something like: A strength of this study is the use of two separate participant cohorts with different viewpoints (personal and professional).

We have amended as below:

'A strength of this study is the use of two separate participant cohorts with different viewpoints (personal and professional).'

Page 14, line 53

"However, patient desires need to be balanced with HCP requirements as if

the device was designed in a way that disrupted the treatment it would be unlikely to be implemented."

Please consider rewording to improve flow.

We have amended as below:

'However, for the device to be successful it needs to be designed to meet the needs of the patient user and the requirements of minimal disruption to treatment delivery of the HCP, and as such further iterations require both patient and HCP involvement.'

Page 14, line 56

Further qualitative research could be conducted on this topic with the use of

a patient and public involvement project advisory group at all stages of the study to make sure the interests of both groups are equally represented (29, 40).

Repetition – consumer involvement has already been mentioned earlier in the discussion.

We have deleted the relevant section.